# The Effect of Polyphenol Extract from Rosa Roxburghii Fruit on Plasma Metabolome and Gut Microbiota in Type 2 Diabetic Mice

**DOI:** 10.3390/foods11121747

**Published:** 2022-06-14

**Authors:** Hui Wang, Zhaojun Chen, Mei Wang, Mingxiu Long, Tingyuan Ren, Chao Chen, Xiaotong Dai, Sheng Yang, Shuming Tan

**Affiliations:** 1Key Laboratory of Plant Resource Conservation and Germplasm Innovation in Mountainous Region (Ministry of Education), College of Life Sciences/Institute of Agro-Bioengineering, Guizhou University, Guiyang 550025, China; wanghui880101@163.com; 2Institute of Biotechnology, Guizhou Academy of Agricultural Sciences, Guiyang 550025, China; chenzhaojun8811@163.com (Z.C.); wm1375898692@163.com (M.W.); 3Guizhou Research Institute of Modern Agricultural Development, Guizhou Academy of Agricultural Sciences, Guiyang 550025, China; longmingshow2021@163.com; 4Key Laboratory of Agricultural and Animal Products Storage and Processing of Guizhou Province, College of Liquor and Food Engineering, Guizhou University, Guiyang 550025, China; tyren@gzu.edu.cn (T.R.); chaochengzu@163.com (C.C.); ogcogcogcogc@163.com (X.D.); ss133yang@yeah.net (S.Y.)

**Keywords:** Rosa roxburghii fruit, type 2 diabetes, plasma metabolome, gut microbiota

## Abstract

Rosa roxburghii fruit is an underutilized functional food abundant in polyphenols. Polyphenols have been proved to have antidiabetic effects. This study investigates the effects of Rosa roxburghii fruit polyphenols extract (RPE) on plasma metabolites and gut microbiota composition in streptozotocin (STZ)- and high-fat diet- induced type 2 diabetes using metabolomics and 16S rRNA gene sequencing. The induced diabetic mice were fed with 400 mg/kg body weight RPE for 8 weeks. RPE demonstrated hypoglycemic, hypolipidemic, and anti-inflammatory effects. Colonic oxidative stress biomarkers were also lowered by RPE. Besides, RPE decreased plasma ceramides and tyrosine levels and increased carnitine and phosphatidylinositols levels, indicating improved insulin resistance, lipid metabolism, and immune response. Furthermore, RPE decreased abundances of *Lachnospiraceae* and *Rikenellaceae* and increased abundances of *Erysipelotrichaceae* and *Faecalibaculum*. Metabolic function prediction of the gut microbiota by PICRUSt demonstrated that RPE downregulated the phosphotransferase system. Taken together, these findings demonstrated that RPE has the potential to prevent type 2 diabetes by regulating the plasma metabolites and gut microbes.

## 1. Introduction

Type 2 diabetes (T2D) is a metabolic disease characterized by hyperglycemia, insulin resistance, and chronic inflammation [1]. Besides genetic predisposition, healthy lifestyles also affect the development of T2D. A high-calorie diet has been considered a risk factor for T2D. Consumption of a high-fat diet, typically in T2D, induces an imbalance of gut microbiota and disruption of the gut barrier integrity [2]. The exogenous substances, such as lipopolysaccharide (LPS), could cross the disrupted gut barrier to the systemic circulation and induce endotoxemia, further aggravating insulin resistance and inflammation in T2D [2]. Meanwhile, oxidative stress status induced by a high-fat diet in T2D has been shown to activate NF-κB (nuclear factor κB) and release its downstream pro-inflammatory cytokines, such as TNF-α (tumor necrosis factor-α) [3]. Healthy food choices have been recognized as an important approach to the management of T2D [4]. Natural products with antidiabetic properties, such as polyphenols from fruits and vegetables, have drawn increasing attention [4].

Polyphenols, a family of secondary metabolites in plants, are natural phytochemical compounds that are recognized to have antidiabetic effects such as antioxidation, anti-inflammation, modulation of the carbohydrate metabolism, improvement of β-cell function, and insulin action [5,6]. The absorption of dietary polyphenols occurs in the stomach and upper intestine, and the absorbed polyphenols later undergo the phase I and phase II metabolism to produce conjugated phenolic compounds [7]. Due to the poor bioavailability of polyphenols, a major part of dietary polyphenols reaches the colon and is metabolized by gut microbiota [7,8]. Polyphenols-gut microbe interactions follow a two-way relationship, where the gut microbiota is shaped by the diet. Polyphenols and polyphenols can be metabolized into simpler metabolites with improved bioavailability and biological effects [7].

Rosa roxburghii fruit mainly grows in the mountains of China’s southwest and central south regions. This fruit has been historically recorded to be used in traditional Chinese medicine for the treatment of diseases, such as hyperlipemia, diarrhea, and enteritis [6]. Rosa roxburghii fruit is an underutilized food rich in a large number of bioactive compounds, for example, polyphenols, polysaccharides, and ascorbic acid [6]. Phenolic acids and flavonoids are the major phenolic compounds in Rosa roxburghii fruit, such as ellagic acid and isoquercitrin [6,9], which have been reported to have antidiabetic effects via modulating insulin resistance, oxidative status, and lipid metabolisms [10,11]. Recently, a study has investigated the antidiabetic effects of polysaccharides extracted from Rosa roxburghii fruit on the diabetes of *db/db* mice [12]. The polysaccharide extract from Rosa roxburghii fruit has hypoglycemic and hypolipidemic effects in *db/db* mice by downregulating key enzymes of hepatic lipid synthesis and gluconeogenesis such as sterol regulatory element-binding protein-1 and glucose-6-phosphatase [12]. 

However, the potential antidiabetic effect of polyphenols’ extract from Rosa roxburghii fruit has not been studied. Therefore, this study aims to evaluate the effect of polyphenols extract from Rosa roxburghii fruit on metabolic parameters, inflammation markers, plasma metabolome, and gut microbiome composition in STZ- and high-fat diet-induced type 2 diabetic mice.

## 2. Materials and Methods

### 2.1. Polyphenols’ Extraction from Rosa Roxbunghii Fruit

Rosa roxbunghii, collected from the Guizhou province in August 2020, China, were provided by Guizhou Hengli Source Natural Biotechnology Ltd. Whole Rosa roxbunghii fruits, including the peel, were dried at 55 °C for 10 h. Polyphenols were extracted with 80% *v*/*v* aqueous ethanol with a liquid–solid ratio of 10:1 (*v*/*w*). The extraction procedure was repeated three times. The resultant extracts obtained from each extraction were combined and filtered, and the ethanol in the extract was removed by vacuum evaporation. The resultant extract was purified with an equilibrated AB-8 macroporos resin (Sigma-Aldrich, Steinheim, Germany). After which, the column was washed with water to elute the water-soluble acids and sugars, and then 80% aqueous ethanol was used to elute polyphenols. The eluent was collected, and ethanol was removed under vacuum rotary evaporation. The resultant Rosa roxburghii fruit polyphenols’ extract (RPE) was freeze-dried and stored at −80 °C.

### 2.2. Nontargeted Metabolic Profiling Analysis of Rosa Roxburghii Fruit Polyphenols’ Extract

Metabolites extraction from RPE and analysis were performed at Biotree Bio-technology (Shanghai, China) [13]. Briefly, metabolites were extracted from 20 mg aliquot of sample using 700 μL of extract solution (methanol/water = 3:1, containing internal standard 2-Chloro-DL-phenylalanine). After the vortex, the sample was homogenized. Afterwards, the sample was extracted overnight at 4 °C on a shaker and then centrifuged at 12,000 × *g* for 15 min at 4 °C. The samples were filtered with 0.22 μm filters before being subjected to the UHPLC separation system separation (SCIEX, Redwood City, CA, USA). The mobile phase A was 0.1% formic acid in water, the mobile phase B was acetonitrile, and the following gradient was used: 2% B (0–0.5 min); 2–50% B (0.5–10 min); 50–95% B (10–11 min); 95% B (11–13 min); 95–2% B (13–13.1 min); 2% B (13.1–15 min). A Sciex QTrap 6500+ (SCIEX, Redwood City, USA) was used for metabolic profiling. Ionization parameters were: IonSpray Voltage: +5500/−4500 V, DP: ± 100 V, Curtain Gas: 35 psi, Ion Source Gas 1:60 psi, Ion Source Gas 2: 60 psi.

### 2.3. Animal Models and Experimental Design

Four-week-old healthy male Kunming mice with body weight of 20 ± 2 g were provided by Henan Skbeth Biotechnology Co., Ltd. (License No.: SCXK 2020-0005, Zhengzhou, China) and maintained at the Animal Housing Unit of Guizhou University (Guiyang, China) under a 12 h light/12 h dark cycle and controlled temperature (23–25 °C). The animal handling protocols were approved by the Guizhou University Animal Ethics Committee (No.: EAE-GZU-2020-P005). After 1 week of adaptation, the mice were randomly divided into two groups and received different diets: normal chow-fed healthy control group (Con, *n* = 8) and the high-fat diet fed group. Mice in the high-fat diet fed group were induced to become type 2 diabetes by injection of streptozotocin (50 mg/kg body weight i.p.) in ice-cold citrate buffer (pH 7.4). The injection was repeated every 3 days for 3 times. Only those animals with fasting plasma glucose greater than 150 mg/dl were included in the study. The animals with diagnosed diabetes were further divided into two groups: the diabetic model group (M group, *n* = 8) and the RPE-treated group (RPE group, *n* = 8). The diabetic mice in the RPE group were additionally fed with 400 mg/kg body weight RPE for 8 weeks. After 8 weeks of treatment, the rats were sacrificed under isoflurane anesthesia after overnight fasting. Colon tissue and colonic content were collected. Plasma samples were collected after centrifuging the blood samples at 3000× *g* for 10 min at 4 °C. The samples were frozen immediately in liquid nitrogen and stored at −80 °C until analyses.

### 2.4. Biochemical Parameters

The oral glucose tolerance test (OGTT) was performed at week 7. After fasting for 18 h, mice were fed with 1.5 mg/g body weight glucose solution. Blood was withdrawn after snipping the end of the tail, and glucose level was measured at 0, 30, 60, 90, and 120 min, respectively. Glucose levels were monitored and determined with an Accu-chek glycosometer (Roche Diagnostics, Almere, The Netherlands). Plasma glucose, hemoglobin A1c (HbA1c), total cholesterol (TC), total glyceride (TG), low-density lipoprotein cholesterol (LDL-C), and high-density lipoprotein cholesterol (HDL-C) were analyzed by a biochemical analyzer (Mindray BS-480, Shenzhen, China). Levels of superoxide dismutase (SOD), malondialdehyde (MDA), and glutathione peroxidase (GPx) were measured by a determination kit (Wanleibio, Shenyang, China). Insulin in plasma was measured using an ELISA kit (Alpco, Salem, MA, USA) according to the manufactural protocol. Insulin resistance level was calculated based on the homeostatic model assessment of insulin resistance (HOMA-IR) index. HOMA-IR index was calculated by the formula: fasting insulin (μM/mL) × fasting glucose (mM)/22.5. The levels of TNF-α and IL-1β in plasma and the colon were determined by ELISA kits (Nanjing Jiancheng Bioengineering Institute, Nanjing, China). The plasma samples were diluted according to the ELISA instruction. The tissue samples were homogenized with cold phosphate buffered saline, and the resultant homogenates were centrifuged at 4 °C for 5 min at 8000× *g*; the supernatant was used for ELISA analysis.

### 2.5. Plasma Non-Targeted Metabolomics Analysis

A total of 24 plasma samples were included in this study (*n* = 8, each group). A plasma nontargeted metabolomics analysis was performed at the Biotree Bio-technology (Shanghai, China) using a UHPLC system (Vanquish, Thermo Fisher Scientific, Waltham, MA, USA) coupled to Orbitrap Exploris 120 mass spectrometer (Orbitrap MS, Thermo). Plasma metabolites were extracted from the blood sample (100 μL) by using 300 μL of methanol containing an isotopically labeled internal standard mixture. Then, the sample was centrifuged at 12,000× *g* for 15 min at 4 °C. The supernatant was collected and filtered for analysis. The quality control (QC) samples were prepared by pooling 10 μL of each sample. The mobile phase A is 5 mmol/L ammonium acetate and 5 mmol/L acetic acid in water and the mobile phase B is acetonitrile. The auto-sampler temperature was 4 °C and the injection volume was 2 μL. The ESI source parameters were capillary temperature 320 °C, spray voltage 3.8 kV (positive) or −3.4 kV (negative), and collision energy as 10/30/60 in NCE mode. 

ProteoWizard was used to convert the raw data to the mzXML files, after which the files were processed with an in-house R program and XCMS for peak picking, peak extraction, peak alignment, retention time correction, and peak integration [14]. Peaks with standard deviations greater than 30% in the QC samples were removed. Metabolite annotation was applied by using an in-house MS2 database (BiotreeDB), which was established with available authentic standards. The cutoff for annotation was set at 0.3. The normalized peak intensities of the metabolites were compared between different groups. *T*-test *p* < 0.05 and VIP values > 1.0 were used to identify significantly differential metabolites. The VIP values for metabolites were calculated using PLS-DA analysis. A two-side hypergeometric test was used as the statistical test method and Benjamini−Hochberg was used as the FDR correction method in the metabolites KEGG enrichment analysis.

### 2.6. Analysis of Gut Microbiota

A total of 24 colonic content were included in this study (*n* = 8, each group). The genomic DNA extraction from colonic content, PCR amplification, and sequencing were conducted by Majorbio (Shanghai, China) and followed the previous protocol described [14]. The V3−V4 hypervariable region of the bacterial 16S rRNA gene was amplified with primers 338F (5′-ACTCCTACGGGAGGCAGCAG-3′) and 806R (5′-GGACTACHVGGGTWTCTAAT-3′). Briefly, resulting PCR products were extracted from PCR amplification by using a 2% agarose gel and further purified using the AxyPrep DNA Gel Extraction Kit (Axygen Biosciences, Union City, CA, USA) and quantified using QuantiFluor™-ST (Promega, Madison, WI, USA). The DNA library was constructed using the TruSeq DNA Sample Prep kit (Illumina, San Diego, CA, USA) and sequenced on an Illumina MiSeq platform (Illumina, San Diego, CA, USA). After sequencing, barcode and primer sequences were removed. Specific tags were generated and cleaned by FLASH software and Trimmomatic software (v0.33). UCHIME software (v4.2) was used to filter chimeric sequences in the obtained clean tags. The resulting sequences with an identity greater than 97% were classified as operational taxonomic units (OTUs). Taxonomic annotation was performed by using the SSU rRNA database. The relative abundance was compared by the Student *t*-test. PICRUSt2 was used to perform microbial metabolic function prediction from 16S rRNA results [15].

### 2.7. Statistical Analysis

The values were defined as the mean ± standard deviation (SD). The differences between the two groups were calculated using the Student’s *t*-test using GraphPad Prism 9. The significance of differences was described as * *p* < 0.05, ** *p* < 0.01, and *** *p* < 0.001 compared with the M group, or # *p* < 0.05, ## *p* < 0.01, and ### *p* < 0.001 compared with the Con group.

## 3. Results

### 3.1. Compound Composition Analysis of RPE Based on Nontargeted Metabolic Profiling

The nontargeted metabolic profiling analysis of Rosa roxburghii fruit polyphenols extract was performed. The composition and content of the Rosa roxburghii fruit polyphenol extract was shown in Appendix A. The polyphenols’ content is 37.96% (including phenols, flavonoid, stilbenes, and lignans), in which isoquercitrin (13.49%) and cianidanol (11.31%) were the most abundant polyphenols.

### 3.2. Effects of RPE on Lipid and Glucose Metabolism Parameters and Oxidative Stress Markers

The body weight of rats in the M group started to be significantly higher than that of the Con group on the 14th day of the intervention and RPE prevented the increased body weight in the M group (Figure 1A). Liver weight was also decreased by RPE (Figure 1B). Plasma parameters demonstrated the abnormal lipid metabolism, glucose metabolism, and immune responses in the M group. Compared to the Con group, the M group demonstrated higher levels of LDL, total cholesterol, and triglyceride (Figure 1C,D,F). HDL level was lower in the M group compared to the Con group (Figure 1E). Fasting blood glucose, the glucose level in the OGTT test, hemoglobin A1c (HbA1c), insulin, and HOMA-IR levels were higher in the M group compared to the Con group (Figure 1G–K). The plasma IL-β level was higher in the M group compared to the Con group (Figure 1M). No difference was observed in TNF-α between the M and Con groups (Figure 1L). Administration of RPE significantly decreased fasting blood glucose, HbA1c, insulin, HOMA-IR, LDL, total cholesterol, triglyceride, plasma IL-β, and plasma TNF-α levels and increased the HDL level as well as improved glucose intolerance. 

Colon length was longer in the Con group compared to the M group, whereas, RPE could prevent the decrease in the colon length in diabetes (Figure 2A). STZ- and high-fat diet-induced diabetes also demonstrated abnormal immune responses in the colon with the increased levels of colonic IL-β and TNF-α and the RPE decreasing these parameters (Figure 1B,C). Colonic oxidative stress markers were evaluated, superoxide dismutase (SOD), glutathione peroxidase (GPx), and malondialdehyde (MDA) were significantly altered in the M group compared to the Con group and RPE significantly reversed these oxidative stress markers to the normal levels (Figure 1D,E).

### 3.3. Effects of RPE on the Plasma Metabolites

Venn diagrams were constructed to show overlaps in significantly upregulated and downregulated metabolites between comparisons (M/Con and M/RPE), showing the levels of the metabolites restored by RPE in diabetic mice. A total of 252 metabolites were increased in the M group compared to the Con group, in which 14 metabolites were decreased by RPE (Figure 3A,C). A total of 230 metabolites were decreased in the M group compared to the Con group and 22 of which were increased by RPE (Figure 3B,D). Pathway enrichment analyses were conducted using the Metaboanalyst database. Increased metabolites in the M group compared to the Con group were enriched in the glutathione metabolism, fructose and mannose metabolism, amino sugar and nucleotide sugar metabolism, alanine, aspartate, and glutamate metabolism pathways (Figure 3E). Increased metabolites in the M group compared to the RPE group were enriched in steroid hormone biosynthesis (Figure 3F). Decreased metabolites in the M group compared to the Con group were enriched in the alpha-linolenic acid metabolism, tryptophan metabolism, arginine and proline metabolism, and amino sugar and nucleotide sugar metabolism (Figure 3G). Decreased metabolites in the M group compared to the RPE group were enriched in sphingolipid metabolism and tyrosine metabolism (Figure 3H).

### 3.4. Effects of RPE on the Gut Microbiota Composition and Function

To analyze the altered gut microbial composition in diabetes induced by STZ and a high-fat diet and the effect of the R. roxburghii fruit polyphenol extract (RPE), a metagenomic DNA sequencing was performed. The PCoA analysis revealed distinguished gut microbiota composition profiles among groups (Figure 4A). α-Diversity indexes representing the richness and evenness of gut microbiota diversity, including Shannon, Chao, and Ace, did not show the difference between the diabetic M group and Con group; however, the Shannon index was increased in the RPE group compared to the diabetic and Con groups (Figure 4B–D). Fecal microbial composition at the phylum level is shown in Figure 4E,F. The abundances of Firmicutes, Bacteroidota, Desulfobacterota, and Proteobacteria showed differences in the M group compared to the Con group. The diabetic M group showed a higher level of Firmicutes/Bacteroidota ratio and RPE reversed the increased Firmicutes/Bacteroidota ratio in the M group to the level of the Con group (Figure 4G). RPE increased the abundance of Desulfobacterota. The abundance of Proteobacteria, which was higher in the Con group compared to the M group, was not affected by RPE. 

At the family level, the top fifteen most abundant families are shown (Figure 5A,B). In the diabetic mice, the relative abundances of *Erysipelotrichaceae*, *Peptostretococcaceae*, and *Clostridiaceae* were increased whereas the abundances of *Lactobacillaceae*, *Muribaculaceae*, *Bifidobacteriaceae*, and *Clostridium* were decreased compared to the Con group. The RPE diet prevented the decreased abundances of *Erysipelotrichaceae* and *Peptostretococcaceae* in diabetes. Moreover, RPE supplementation increased the abundance of *Marinifilaceae* in diabetic animals. Figure 5C shows the top fifteen most abundant genus and Figure 5D shows the seven identified known species from the top fifteen most abundant species. The relative abundances of *Dubosiella*, *Faecalibaculum* (mainly *F. rodentium*), *Romboutsia* (mainly *R. ilealis*), *Enterorhabdus*, *Clostridium,* and *Colidextribacter* were increased whereas the abundances of *Bifidobacterium* (mainly *B. pseudolongum*) and *Desulfovibrio* were decreased in the M group compared to the Con group. The RPE-supplemented diet reversed the increases in the abundances of *Dubosiella*, *Faecalibaculum*, and *Romboutsia* in the M group. Moreover, the increased *Lactobacillus reuteri* in the M group compared to the Con group was decreased by RPE. In addition, RPE also increased the abundance of *Colidextribacter*, *Rikenella* (mainly *R. microfusus DSM15922*), *Desulfovibrio*, and *Alistipes* (*A. inops*) and decreased *Lactobacillus* in diabetic animals. 

The most different top thirty KEGG pathways generated from M/Con and M/RPE comparisons are shown in Figure 6A,B. The top three most upregulated pathways in the diabetic M group compared to the healthy Con group were transporters, ABC transporters, and bacteria motility proteins. The top three most downregulated pathways in the diabetic M group were ribosome, other glycan degradation, and purine metabolism. RPE upregulated the phosphotransferase system, amino sugar, and nucleotide sugar metabolism, and fructose and mannose metabolism as the top three most upregulated pathways in diabetes. Energy metabolism, valine, leucine, and isoleucine biosynthesis, as well as the citrate cycle, were downregulated by RPE in diabetes.

## 4. Discussion

Rosa roxburghii fruits have drawn much attention due to their high abundance of bioactive compounds, such as polyphenols, superoxide dismutase, polysaccharide, vitamin C, and vitamin E [16]. Water-soluble polysaccharide extract from Rosa roxburghii fruits has demonstrated the potential to be a hypoglycemic agent [12]. This is the first study investigating the impact of Rosa roxburghii fruit polyphenols’ extract on plasma metabolites and the composition of gut microbiota in diabetic mice. Different plasma metabolic profiles and gut microbiota between diabetic mice and healthy rats and the effects of Rosa roxburghii fruit polyphenols’ extract were observed.

T2D is a serious disease characterized by disturbed lipid and glucose metabolism as well as inflammatory response disorders [17]. Therefore, the regulation of lipid and glucose metabolism and inflammation is one of the strategies for the treatment and management of T2D. In this study, the RPE reversed the augmentation of the body weight and liver weight in diabetic mice. Plasma lipid profile and glucose were also improved by the extract. Surprisingly, the increased plasma insulin and insulin resistance index HOMA-IR were reversed by the RPE extract to the normal levels in diabetic mice. Low-grade and chronic inflammation has been considered a contributor to the pathogenesis of type 2 diabetes [18]. The RPE decreased pro-inflammatory cytokines IL-β and TNF-α in plasma and colon tissue. The results were consistent with previous studies, which have reported that the intake of polyphenol extracts from purple sweet potato [19] and black goji berry [20] have improved the inflammatory response in diabetes. Gut inflammation and oxidative stress have been identified in diabetes [21,22,23]. Oxidative stress has been considered one of the risk factors in the development of diabetes [24]. Oxidative stress emerges when free radical production exceeds the antioxidant capacity. The free radicals, such as reactive oxygen species, can disrupt cellular compounds and convert membrane lipoproteins to pro-inflammatory substances [25]. Free radicals could initiate the lipid peroxidation, by which malondialdehyde (MDA), as one of the final products of polyunsaturated fatty acids of peroxidation, is produced [26]. Antioxidant enzymes such as glutathione peroxidase (GPx) and superoxide dismutase (SOD) have free radical scavenging properties [26]. In this study, increased colonic MDA in the M group was significantly decreased by RPE; the extract also increased colonic antioxidant enzymes, SOD and GPx. These results indicated the Rosa roxburghii fruit polyphenols’ extract improved the inflammation as well as the oxidative stress in the colon.

The M group demonstrated dysregulated carbohydrate and amino acids from the KEGG pathway enrichment analysis. Besides increased glucose level in the plasma, the M group demonstrated upregulated fructose and mannose metabolism, which was in line with a previous diabetes study [27]. Glutathione metabolism was upregulated in the M group and mainly contributed to by increased glutathione; however, glutathione is an antioxidant and has been reported to be inversely correlated with fasting glycemia in patients with type 2 diabetes [28], which was contrary to the result in this study and deserved further investigation. The relationship between steroid metabolism and diabetes has been established. Increased steroids have been associated with insulin resistance [29]. Upregulated steroid metabolism was also observed in the M group. Rosa roxburghii fruit polyphenols extract downregulated steroid metabolism, suggesting improved insulin resistance. In addition, RPE upregulated sphingolipid metabolism, which is mainly contributed by the increased sphinganine 1-phosphate in the RPE group

Ceramides have been reported to be able to decrease insulin sensitivity in the liver, muscle, and adipose tissue due to their ability to desensitize leptin [30]. Ceramide might also contribute to cardiovascular diseases [30], since it has been reported to increase atherosclerotic plaques and inflammatory arteriolar dilation. Preventing the accumulation of ceramides can prevent insulin resistance, cardiovascular diseases, and hepatic steatosis [31,32]. Two ceramides, Cer/AP(t15:2/16:2) and Cer/AS(d15:2/16:2), were increased in the M group while RPE decreased these levels, indicating that the RPE might improve the sensitivity of leptin and insulin resistance in diabetes. Circulating tyrosine is highly associated with insulin resistance in diabetes [33]. The mechanism might be that tyrosine aminotransferase can catalyze tyrosine to p-hydroxyphenylpyruvate and thus lower the tyrosine level. Hyperinsulinemia as a result of insulin resistance in diabetes can activate the tyrosine aminotransferase [34]. In this study, the M group demonstrated increased tyrosine level and RPE decreased its level to the normal state, contributing to the improved insulin resistance by RPE. Three glucuronidated products, estriol-17-glucuronide, retinyl beta-glucuronide, and tetrahydroaldosterone-3-glucuronide, were increased in the M group, which have been also observed in patients with metabolic syndrome [35]. The increased glucuronidated products indicated an altered glucuronidation process, which occurs primarily in the liver [36]. RPE decreased these products, suggesting a regulation of glucuronidation in the liver by polyphenols. L-acetylcarnitine and O-acetylcarnitine were acetylated derivatives of the amino acid L-carnitine. L-Carnitine is a key substance in regulating lipid metabolism and fatty acid beta-oxidation. It facilitates transporting long-chain fatty acids into the mitochondria from the cytosol to be oxidized for energy production [37]. L- acetylcarnitine transports acetyl-CoA to mitochondria during the oxidation of fatty acids [37]. It is well established that lipid oversupply or lipotoxicity affects insulin resistance in diabetes; for example, the accumulating fatty acyl CoA-derived substances dampen insulin signaling and glucose oxidation by activating a serine/threonine kinase cascade and eventually causing the dysregulation of insulin receptors [38]. L- Acetylcarnitine, L-carnitine, and O-acetylcarnitine were decreased in the M group, indicating a possible accumulation of lipid and lipotoxicity due to fewer fatty acids being transported to mitochondria and being oxidized. Rosa roxburghii fruit polyphenols extract increased L-carnitine, L- acetylcarnitine, and O-acetylcarnitine levels and possibly restored lipid metabolism. Phosphatidylinositols are the primary source of the arachidonic acid, which is a key intermediate for the biosynthesis of eicosanoids [39], and eicosanoids participate in modulating the intensity and duration of immune responses [40]. Ten phosphatidylinositols were decreased in the diabetic mice and RPE increased these phosphatidylinositols, indicating a potential modulatory effect on immune responses by RPE.

The 16S rRNA gene was sequenced to study the effects of RPE on the gut microbiota. The RPE increased the α-Diversity indexes, although the significant change was only observed in the Shannon index. Firmicutes/Bacteroidota ratio has been proposed as a biomarker for obesity and type 2 diabetes [41]. In this study, the M group demonstrated a higher Firmicutes/Bacteroidota ratio compared to the Con and RPE decreasing this ratio, indicating a positive effect on gut microbiota by RPE. A higher level of *Erysipelotrichaceae* has been found in obesity and inflammation-related gastrointestinal diseases [42]. In this study, RPE decreased the high abundance of *Erysipelotrichaceae* in the M group. Moreover, RPE increased the *Lachnospiraceae* family, which has been reported to be one of the major propionate-producing bacteria in gut flora [43]. The *Lachnospiraceae* family has also been reported to be increased by phenolic extracts from bilberries and purple potatoes [44]. The family *Rikenellaceae* has been recognized as protective bacteria in colitis and are inversely associated with pro-inflammatory cytokine IL-18 [45]. However, alloxan-induced diabetes has demonstrated an increased Family *Rikenellaceae* abundance [46]. In this study, there was no significant change in the abundance of *Rikenellaceae* between the STZ- and high-fat diet-induced diabetes and the healthy controls; however, RPE increased *Rikenellaceae* including its genus *Rikenella* (mainly *R. microfusus DSM15922* species) and *Alistipes* (mainly *A. inops* species). *Faecalibaculum* has been found to produce lactate, and a high level of lactate has also been reported in human subjects with T2D and obesity [47]. STZ- and high-fat diet-induced diabetes demonstrated an increased level of *Faecalibaculum* (mainly contributed by *F. rodentium*) compared to the healthy controls, and RPE decreased the abundance of *Faecalibaculum*. The reduced abundance of *Bifidobacterium* spp., including *B. psedolingum,* have been reported to be associated with the T2D [48]; in this study, the M group demonstrated a lower abundance of *Bifidobacterium**;* however, the RPE showed little effect on it. One of the most important findings in the microbial metabolic function revealed by PICRUSt2 was that the M group had a more active phosphotransferase system (PTS) pathway compared to the Con group and RPE decreased the activity of PTS. PTS has been found to be more active in obese children [49] and patients with diabetes [50]. PTS controls carbohydrate uptake and the phosphorylation of carbohydrates; PTS also modulates gene expression involved in the carbohydrate metabolism [49,51]. Furthermore, PTS has been found to be primarily present in Actinobacteria and Firmicutes [49] and RPE might downregulate the PTS pathway by decreasing the abundance of Actinobacteria and Firmicutes.

## 5. Conclusions

Taken together, this study revealed the modulatory effects of Rosa roxburghii fruit polyphenols’ extract on hyperglycemic, hyperlipidemic, and inflammatory cytokines as well as plasma metabolome and gut microbiota in STZ- and high-fat diet-induced diabetes. Rosa roxburghii fruit polyphenols’ extract decreased plasma glucose, plasma inflammatory cytokines, and insulin resistance and improved the plasma lipid profile. Oxidative stress markers and inflammatory cytokines in the colon were also decreased by Rosa roxburghii fruit polyphenols’ extract. The extract modulated plasma metabolites, for example, ceramides and tyrosine, were decreased and carnitine and phosphatidylinositols were increased by the extract, indicating improved insulin resistance, lipid metabolism, and immune response. Rosa roxburghii fruit polyphenols extract also modulated gut microbes, such as *Lachnospiraceae* and *Rikenellaceae,* with beneficial effects, were increased and *Erysipelotrichaceae* and *Faecalibaculum* having a negative role in diabetes were decreased by the extract. Although the antidiabetic effect of Rosa roxburghii fruit polyphenols’ extract was revealed in this study, additional work is required to investigate the in-depth mechanism of the antidiabetic effect of the extract; for example, a fecal microbiota transplant can be used to further investigate whether the altered gut microbiota is a key contributor to the antidiabetic effect of the extract. Overall, this is the first study discovering the anti-diabetic effects of the Rosa roxburghii fruit polyphenols’ extract, which promotes the incorporation of Rosa roxburghii fruit or its polyphenols’ extract into the daily diet to prevent or ameliorate diabetes.

## Figures and Tables

**Figure 1 foods-11-01747-f001:**
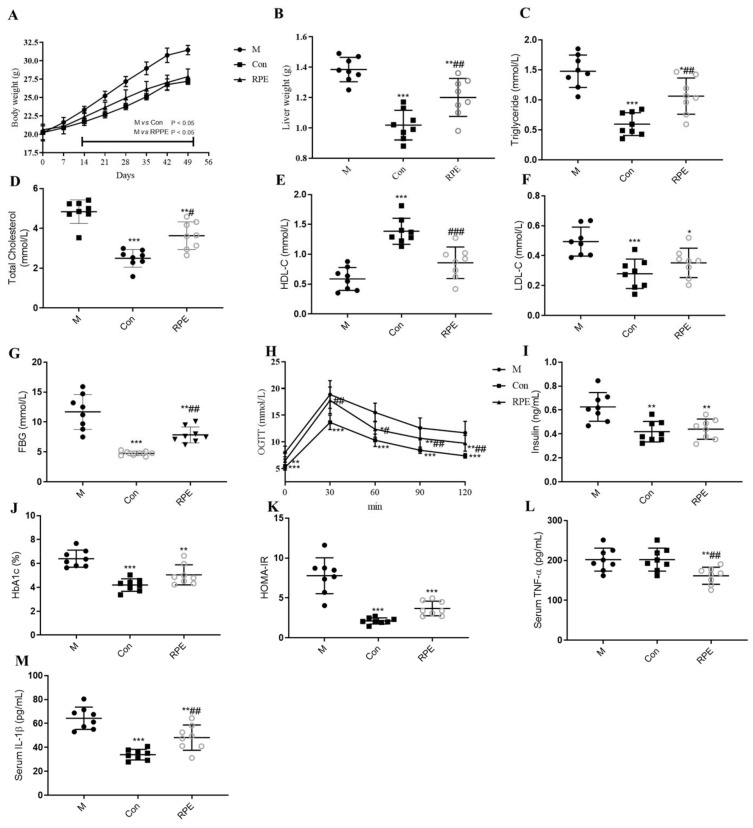
Effects of Rosa roxburghii fruit polyphenols extract on the bodyweight (**A**), liver weight (**B**), plasma triglyceride (**C**), plasma total cholesterol (**D**), HDL-C (**E**), LDL-C (**F**), fasting blood glucose (FBG) (**G**), oral glucose tolerance text (**H**), insulin (**I**), HbA1c (**J**), HOMA-IR (**K**), plasma TNF-α (**L**), and plasma IL-β (**M**) in STZ-induced diabetic mice. Note: * *p* < 0.05, ** *p* < 0.01, and *** *p* < 0.001 versus the M group; # *p* < 0.05, ## *p* < 0.01, and ### *p* < 0.001 vs. the Con group.

**Figure 2 foods-11-01747-f002:**
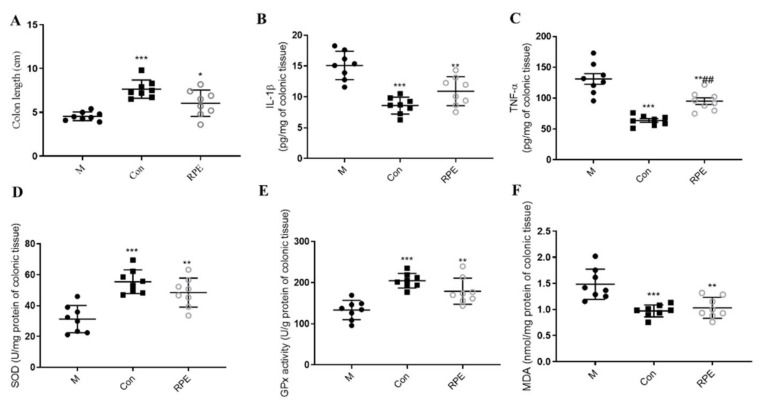
Effects of Rosa roxburghii fruit polyphenols extract on the colon length (**A**), colonic IL-β (**B**), colonic TNF-α (**C**), colonic superoxide dismutase (SOD) (**D**), colonic glutathione peroxidase (GPx) (**E**), and colonic malondialdehyde (MDA) (**F**) in STZ-induced diabetic mice. Note: * *p* < 0.05, ** *p* < 0.01, and *** *p* < 0.001 versus the M group; ## *p* < 0.01, versus the Con group.

**Figure 3 foods-11-01747-f003:**
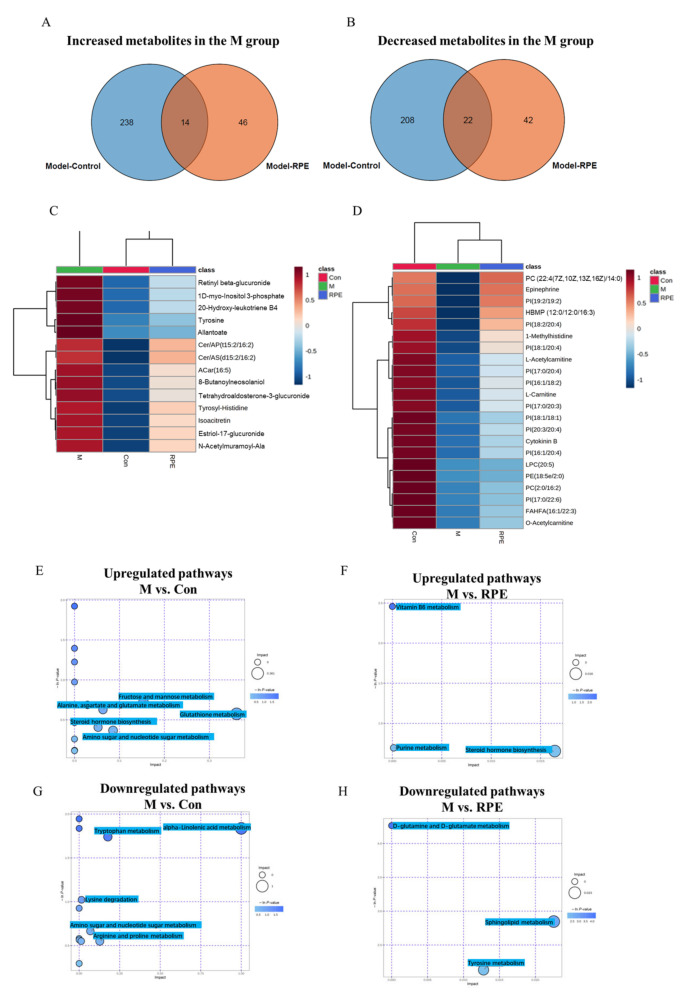
Venn diagrams showing the number of increased metabolites in the M group compared to the Con groups were decreased by RPE (**A**) and the number of decreased metabolites in the M group compared to the Con groups were increased by RPE (**B**). Increased metabolites in diabetes that were reversed by RPE (**C**). Decreased metabolites in diabetes that were reversed by RPE (**D**). KEGG pathway enrichment analysis based on increased metabolites in M/Con (**E**) and M/RPE (**F**) comparisons. KEGG pathway enrichment analysis based on decreased metabolites in M/Con (**G**) and M/RPE (**H**) comparisons.

**Figure 4 foods-11-01747-f004:**
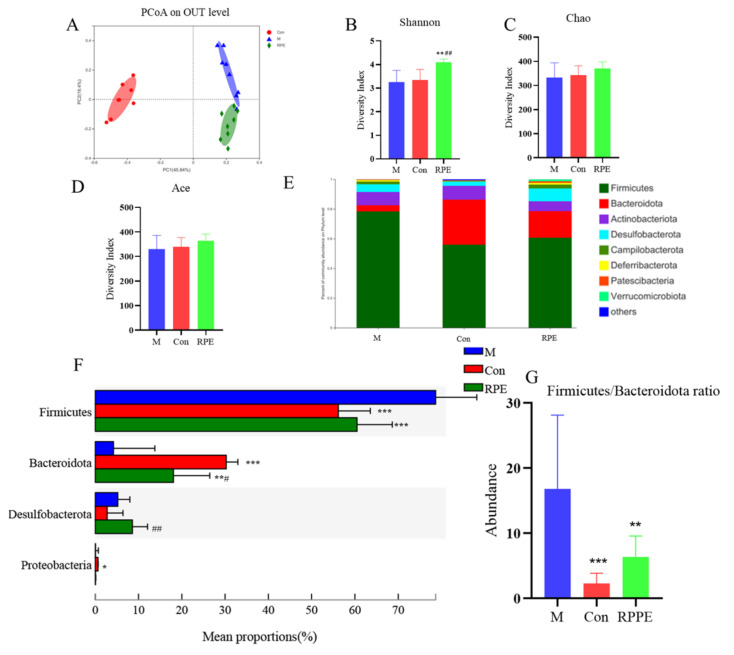
PCoA plot of gut microbiota at genus level (**A**). Shannon index (**B**). Chao index (**C**). Ace index (**D**). The components (**E**) and comparisons (**F**) of gut microbiota at phylum level. Firmicutes/Bacteroidota ratio (**G**) Note: * *p* < 0.05, ** *p* < 0.01, and *** *p* < 0.001 versus the M group; # *p* < 0.05, ## *p* < 0.01, and vs. the Con group.

**Figure 5 foods-11-01747-f005:**
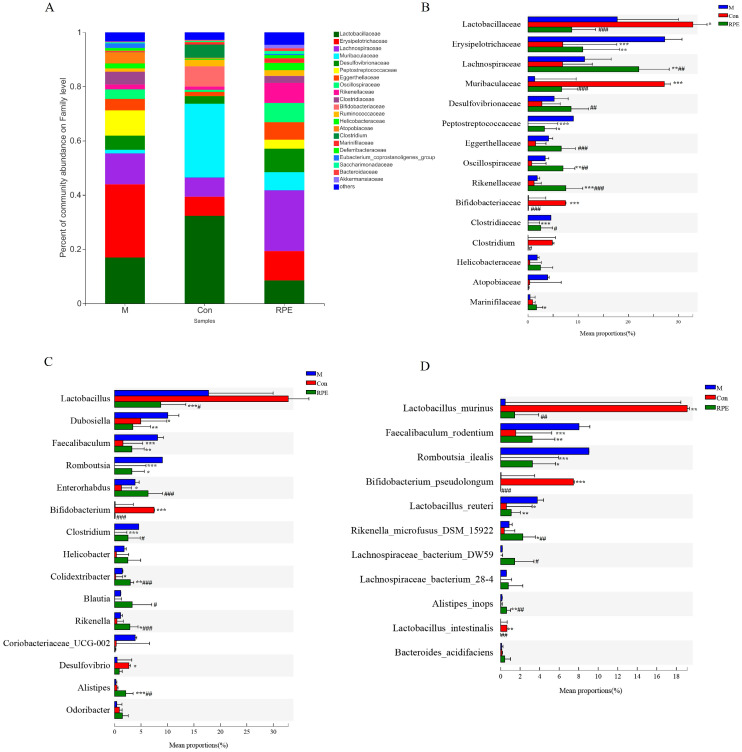
Effects of Rosa roxburghii fruit polyphenols extract on gut microbiota at family (**A**,**B**), genus (**C**), and species (**D**) level. Note: * *p* < 0.05, ** *p* < 0.01, and *** *p* < 0.001 versus the M group; # *p* < 0.05, ## *p* < 0.01, and ### *p* < 0.001 versus the Con group.

**Figure 6 foods-11-01747-f006:**
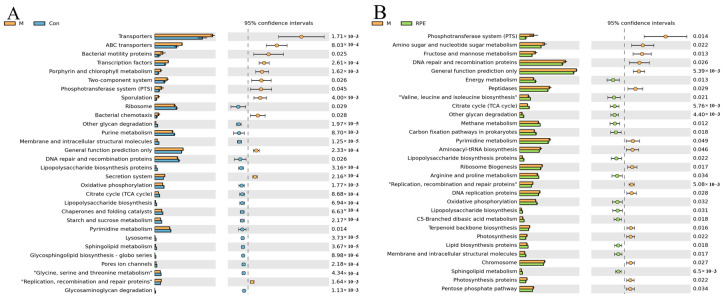
Function prediction of the gut microbiota revealed by PICRUSt2. M group versus Con group (**A**) or RPE group (**B**). Note: *p* < 0.05, confidence intervals = 95%.

## Data Availability

Data is contained within the article and Appendix A.

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
