# Peer review of "The Effect of Polyphenol Extract from Rosa Roxburghii Fruit on Plasma Metabolome and Gut Microbiota in Type 2 Diabetic Mice"

_foods, 2022, doi:10.3390/foods11121747_

Round 1

Reviewer 1 Report

The paper by Zang and colleagues describes the effect of Rosa roxburghii polyphenol in type 2 diabetes patients by inspecting plasma metabolome and gut microbiota i

The paper is well written and the whole workflow followed is in line with similar published papers. Anyway the statistical method description can be improved.

- In the whole paper a better description of the applied statistics is required.

- Please describe the denoising steps and provide a table including the relative statistics and the whole bioinformatics protocol. No details have been provide on bioinformatics pipeline used. 

- I suggest to include a paragraph dealing with bioinformatics.

- A table reporting relative OTU percentage is also missed.

- Line 225 . Here it is not clear if in the M/Con comparison you are referring only to significative metabolites or not. I suppose that the Pathway enrichment conducted using the Metaboanalyst database included a statistical comparison. Please describe whether a correction for multiple tests has been conducted or not.

- Figure 4. Please add a weight estimation of the plotted PCA plot, like the cos2.

Author Response

The paper by Zang and colleagues describes the effect of Rosa roxburghii polyphenol in type 2 diabetes patients by inspecting plasma metabolome and gut microbiota i

The paper is well written and the whole workflow followed is in line with similar published papers. Anyway, the statistical method description can be improved.

We thank the reviewer for reviewing this manuscript and giving positive feedbacks and useful comments.

- In the whole paper a better description of the applied statistics is required.

Description of the applied statistics was added to Methods Line 167-170 Line 187-188

- Please describe the denoising steps and provide a table including the relative statistics and the whole bioinformatics protocol. No details have been provide on bioinformatics pipeline used.

Relative statistics and the whole bioinformatics protocol were added to the Method. Line 160-170 Line 178-188

- I suggest including a paragraph dealing with bioinformatics.

The steps of bioinformatics was added Line 160-170 Line 178-188

- A table reporting relative OTU percentage is also missed.

Our OTU data is needed for other data analysis, for example, the pomace of the fruits has also been investigated to affect the gut microbiota composition, these two data sets will be combined and subjected to other data analysis. We will publish the data once we have done that.

- Line 225 . Here it is not clear if in the M/Con comparison you are referring only to significative metabolites or not. I suppose that the Pathway enrichment conducted using the Metaboanalyst database included a statistical comparison. Please describe whether a correction for multiple tests has been conducted or not.

Those upregulated and downregulated metabolites are significantly changed metabolites between two groups. The significantly changed metabolites were determined by a student T-test. The determination of significantly changed metabolites was added to the Method Line 167.  For pathway enrichment, A two-side hypergeometric test was used as the statistical test method and Benja-mini−Hochberg was used as the FDR correction method in the metabolites KEGG enrichment analysis. The missing description was added to the Method Line 167-170.

- Figure 4. Please add a weight estimation of the plotted PCA plot, like the cos2.

Thank the reviewer for the useful comment. We agree that weight estimation can provide information about variables contribution, however, the PCA plot here was used to just display the distinguished gut microbiota composition profile. The key point of this part is to reveal the microbes that are affected by our RPE treatment. The differences of microbes at different taxonomic levels between M and RPE groups has been revealed by a t-test in Figure 4 and 5 which would be enough to identify the changed microbes, weight estimation of the PCA plot is not as important as the results of Figure 4 and 5 regarding the research question here.

Reviewer 2 Report

The study reports the beneficial effects of polyphenolic extract from Rosa roxburghii fruits in diabetic mice. Few concerns need to be addressed:

1) The processing of fruits for extraction of polyphenols: Whether the fruits were peeled before the extraction or not? As the peel of the fruits is a rich source of polyphenols

2) Number of replications for each test: The authors used 8 mice per group, which were sacrificed at the end of the study for biochemical parameters. However the authors did not report how they have collected the samples for microbiota analysis? How many samples were taken? Whether the samples were pooled per group? How the results were analysed with respect to original microbial population at the starting point of study?

3) It is very interesting to see that the authors report decrease in Bifidobacteria and Lactobacilli count after RPE treatment which would lead to decrease in population of beneficial bacteria. 

4) It should be Venn diagram?

5) It should be Bacteroidetes

Author Response

The study reports the beneficial effects of polyphenolic extract from Rosa roxburghii fruits in diabetic mice. Few concerns need to be addressed:

We thank the reviewer for providing us with useful comments.

1) The processing of fruits for extraction of polyphenols: Whether the fruits were peeled before the extraction or not? As the peel of the fruits is a rich source of polyphenols

We agree with the reviewer that the peel of the fruits is rich in polyphenols. In this study, whole fruits including peels were under extraction process. This was further clarified in the Methods. L83

2) Number of replications for each test: The authors used 8 mice per group, which were sacrificed at the end of the study for biochemical parameters. However, the authors did not report how they have collected the samples for microbiota analysis? How many samples were taken? Whether the samples were pooled per group? How the results were analysed with respect to original microbial population at the starting point of study?

8 mice were used in each group of this study. All replicates (n=8) were used for biochemical parameter analysis, plasma metabolomics as well as gut microbiota analysis. The description was added in the Method. Line 147 Line 173. The samples were not pooled and each of the samples represented one individual mice.

Although the original microbial population at the starting point of the study was not investigated, we agree that it could have been better if we have tested the original microbial population at the starting point to make sure the gut microbiota changes were induced by the intervention, not by improper grouping method. Nevertheless, the mice were randomly divided at the starting point of the study Line 113, which should ensure consistency of gut microbiota profile among these groups at the starting point of the study.

3) It is very interesting to see that the authors report decrease in Bifidobacteria and Lactobacilli count after RPE treatment which would lead to decrease in population of beneficial bacteria.

RPE decreased Lactobacillus in diabetic animals and showed little effect on Bifidobacterium. The change of Lactobacillus was mainly contributed by the species Lactobacillus murinus which was not significantly decreased by RPE. We agree with reviewer that this led to a decrease in the population of beneficial bacteria. However, RPE also modulated other gut microbes, such as Lach-nospiraceae and Rikenellaceae with beneficial effects that were increased and Erysipelotrichaceae and Faecalibaculum having a negative role in diabetes were decreased by the extract.

4) It should be Venn diagram?

Typo was checked and changed to Venn diagram Line 240

5) It should be Bacteroidetes

The phylum Bacteroidota and Bacteroidetes are synonyms for each other. The term “Bacteroidota” has also been used in MDPI journal and others1,2.

(1)        Brzoska, R. M.; Edelmann, R. E.; Bollmann, A. Physiological and Genomic Characterization of Two Novel Bacteroidota Strains Asinibacterium Spp. OR43 and OR53. Bact. 2022, Vol. 1, Pages 33-47 2022, 1 (1), 33–47.

(2)        Duca, S. Del; Riccardi, C.; Vassallo, A.; Fontana, G.; Castronovo, L. M.; Chioccioli, S.; Fani, R. The Histidine Biosynthetic Genes in the Superphylum Bacteroidota-Rhodothermota-Balneolota-Chlorobiota: Insights into the Evolution of Gene Structure and Organization. Microorg. 2021, Vol. 9, Page 1439 2021, 9 (7), 1439.
